# New historical data for long-term swordfish ecological studies in the Mediterranean Sea

Brian R. MacKenzie[1], Teresa Romeo[2,3], Piero Addis[4], Pietro Battaglia[2], Pierpaolo Consoli[2], Franco Andaloro[2], Gianluca Sarà[5]

[1]National Institute for Aquatic Resources, Technical University of Denmark, DK 2800 Lyngby, Denmark

[2]Stazione Zoologica Anton Dohrn, Sicily Marine Centre, Department of Integrative Marine Ecology, Via dei Mille 46, Milazzo (ME), Italy.

[3]National Institute for Environmental Protection and Research,  Via dei Mille 46 - 98057 Milazzo (ME), Italy

[4]Department of Life and Environmental Sciences, University of Cagliari, Sardinia, Italy

[5]Department of Earth and Marine Science, University of Palermo, Italy

Correspondence to: Brian R. MacKenzie (brm@aqua.dtu.dk) and Teresa Romeo (teresa.romeo@szn.it)

**Abstract.** Management of marine fisheries and ecosystems is constrained by knowledge based on datasets with limited temporal coverage. Many populations and ecosystems have been perturbed long before scientific investigations have begun. This situation is particularly acute for the largest and commercially most valuable species.  We hypothesized that historical trap fishery records for bluefin tuna, *Thunnus thynnus* Linnaeus 1758, could contain catch data and information for other, bycatch species, such as swordfish, *Xiphias gladius* Linnaeus 1758. This species has a long history of exploitation and is presently overexploited, yet indicators of its status (biomass) used in fishery management only start in 1950.  Here we examine historical fishery records and logbooks from some of these traps and recovered ca. 110 years of bycatch data (1896-2010). These previously-neglected, but now recovered, data include catch dates and amounts in numbers and/or weights (including individual weights) for the time period before and after major expansion of swordfish fisheries in the Mediterranean Sea. New historical datasets such as these could help understand how human activities and natural variability interact to affect long-term dynamics of this species.

**1 Introduction**

Marine populations, species and ecosystems have been exploited for goods and services for long periods of time before scientific records of this exploitation are available. The written scientific record of changes in such populations and systems can, and does in many cases, underestimate the degree to which society has impacted abundances, sizes, diversity or ecosystem functioning (Fortibuoni et al., 2010; Lotze et al., 2014; MacKenzie et al., 2011; Pauly, 1995). Consequently, sustainability objectives and actions based on short contemporary time perspectives (e.g. 2-3 decades, as is typical for many marine

ecological time series) may be an insufficient basis on which to make management decisions regarding recovery to former states (Caswell et al., 2020; Engelhard et al., 2016; Jackson et al., 2001).

Time series length has other implications for ecological understanding and management, aside from those outlined above. Short time series may be insufficient to allow diagnosis and attribution of the causes of changes in their dynamics. Given that

many of the changes in marine ecosystems are consequences of multiple drivers operating at multi-annual or multi-decadal time scales (Daskalov et al., 2007; Lotze et al., 2014; Piroddi et al., 2017; Reusch et al., 2018), and that many of these changes can overlap in time, a short time series based on 2-3 decades of observations will make it difficult for scientists to resolve which drivers have led to changes. Consider for example, a system (e. g., population, species, food web, ecosystem) which has experienced increases in fishing effort over most of the 20th century; at the same time the system might also have been perturbed

by other drivers such as eutrophication, introduction of non-native species, and increasingly by climate change. A time series lasting 2-3 decades is unlikely to resolve which of these drivers have most influenced the dynamics of the response.

This situation applies to many fish populations in the Mediterranean Sea, where available time series are generally relatively short (Ferretti et al., 2015; Fortibuoni et al., 2010). Furthermore, many of the stocks are overexploited, even based on the

relatively short time series which are available (Cardinale et al., 2017). One such stock is the swordfish, whose available biomass time series starts in 1985 (ICCAT, 2019), and for which officially reported catch data for many countries start only in the 1950s (Spain) or 1960s (e. g., Italy, Greece) (ICCAT, 2020).

This species has however been exploited since antiquity, at least in some parts of the Mediterranean. Harpoon fishermen have

been catching swordfish in the central Mediterranean (Strait of Messina) since millennia (Battaglia et al., 2018; Ward et al., 2000), and some swordfish bones dated from the 4th-10th centuries AD have been found in archaeological sites in Istanbul (Yuncu, 2017). Officially reported catch data for some countries are available since 1950, with reporting levels increasing over time (ICCAT, 2020). The exploitation sharply increased during the 1960s-1980s when new gears and deployment strategies were implemented, including longlines and driftnets. Following these changes, reported landings increased from a few 100

t/year in the 1950s to ca. 15,000 t/year in the early 1980s (ICCAT, 2019). Despite the implementation of several technical measures to preserve the resource, the stock is now currently considered to be overexploited and to have a biomass which is 72% of the long-term sustainable level of biomass which could support maximum fishery yield (i.e., Bmsy) (ICCAT, 2020).

However, the current time series of biomass estimates (1950-2018) and associated fishery-related reference points are

uncertain, particularly before 1985 when fishing effort and other biological data is limited (ICCAT, 2020). As with many other fish stocks, the length of the available biomass and catch time series for Mediterranean swordfish limits understanding of longer-term dynamics and drivers of change (e. g., exploitation, low-frequency climate variations such as the Atlantic Multidecadal Oscillation), which may have initiated before biomass time series begin or which contain too few cycles of variability for detection.  New data sources are needed to improve estimates of historical states and ecological understanding

of long-term dynamics (Damalas et al., 2015). Such data sources could include information about biomass and lifehistory properties such as sizes, growth rates, distributions and migration behaviour.

Here we propose that additional catch data for swordfish from earlier time periods than 1950 may be available in presently overlooked or neglected data sources. One such source could be the trap records from the bluefin tuna fishery. This species

has been caught in coastally-deployed traps for centuries at many locations around the Mediterranean Sea (Addis et al., 2008; Ambrosio and Xandri, 2015; Ravier and Fromentin, 2001). These traps potentially catch other species as bycatch, and in numbers which could provide new ecological insights. Here we provide new data on catch amounts, fishing effort, and individual size which can be used to derive new understanding of swordfish ecology in the Mediterranean Sea and how it is impacted by multiple drivers over multi-decadal time scales. The data recovered here can be used in statistical analyses and

ecological interpretations in new investigations (MacKenzie et al., 2021b).

**2 Methods**

2.1 General description of traps and fishing methods:

Coastally-deployed traps have been used for centuries to capture bluefin tuna while migrating to or from spawning areas in the Mediterranean Sea (Addis et al., 2008, 2012; Ambrosio and Xandri, 2015; Ravier and Fromentin, 2001). Most of these traps are no longer in operation; to our knowledge only one is still operational (Addis et al., 2012). Although the traps were designed to target bluefin tuna, they and other similar smaller traps in the Mediterranean do catch other migratory species

such as smaller tuna species (e. g., bullet tuna, bonito) and elasmobranchs (Britten et al., 2014; Cattaneo-Vietti et al., 2015; Storai et al., 2011). Some reports also mention bycatches of swordfish (Ambrosio and Xandri, 2015; ICCAT, 2020; Rodríguez-Roda, 1964), but the quantities and sizes have so far not been compiled or analysed; consequently the potential value of the traps for generating new insights to long-term dynamcis of swordfish ecology is unknown. Here we hypothesize that these traps also caught swordfish in amounts that can generate new ecological knowledge.

These artisanal bluefin traps have a similar design, which has changed very little over time (Addis et al., 2008, 2012; Ambrosio and Xandri, 2015; Ravier and Fromentin, 2001). This similarity of gear technology in time and space facilitates comparison of catches over time and among locations. Detailed description of the gear design and operation are available in literature (Addis et al., 2008, 2012; Ambrosio and Xandri, 2015). Because of the stable technology, the traps can be considered to be a standardized sampling device within sites (i. e., over time) and among sites (Addis et al., 2012). The

general trap design consists of a long "tail" or fence which guides migrating fish to a series of chambers or rooms, known in Italy as the "castle". The traps were typically deployed by having the long tail portion oriented perpendicular to the shore; the castle portion was at the offshore end of the tail. Water depths at the location of the castle or room portion of the trap were ca. 30-50 m. Differences among areas or over time were generally related to the mouth position of each trap, which was adjusted in relation to several local factors (e. g., bathymetry, and wind and current conditions) and decisions of the trap

manager.

Fish swimming alongshore would then be intercepted and met by the tail of the trap, swim along it and eventually enter the rooms of the castle. The tail was ca. 2 km long, and the mesh size (stretched) was 25 cm (Addis et al., 2008). A visual schematic of the usual deployment position relative to the shore and migration pattern is available in the literature (Addis et al., 2008). Details of the sizes and mesh sizes and deployment locations for the specific traps considered in this study are

summarized in Table 1.

Table 1. Properties and deployment details of bluefin tuna traps considered as sources of swordfish bycatch data in this study.

| Property↓ | Location | | | | |
|---|---|---|---|---|---|
| | Milazzo | Favignana | Formica | Portoscuso | Capo Passero |
| Latitude; longitude | 38 deg., 14 min. N; 15 deg., 13 min. E | 37 deg. 56 min. N; 12 deg., 19 min. E | 37 deg., 59 min. N; 12 deg., 25 min. E | 39 deg., 14 min. N; 8 deg., 22 min. E, | 36 deg., 41 min. N; 15 deg., 9 min. E |
| Number of rooms | 5-7 | 7-8 | 7-8 | 5 | unknown |
| Tail length (m) | 2000 | 1450 | 1200 | 2500 | 1600 |
| Depth at rooms (m) | 30 | 29-32 | 37-41 | 35 | 46-50 |
| Cross orientation | NS | variable | variable | NW-SE | EW |
| Mouth position | SE | variable | variable | SE | variable |
| Mesh size | 25 | - | - | 25 | - |
| Bottom substrate | sand | rock/sand | rock/sand | sand | sand/rock |

Our study is therefore an exploratory data recovery investigation to evaluate the potential that these traps can provide new ecological and fishery-relevant data for a non-target species.  This potential has so far not been widely considered or recognized.

2.2 Data recovery and compilation:

Bluefin tuna traps have been deployed at many locations in Italy, as well as other sites in the Mediterranean Sea and
northeast Atlantic (Morocco, Portugal, Spain) (Ambrosio and Xandri, 2015; Ravier and Fromentin, 2001).  The criteria we used to select trap sites for data recovery included proximity to a major spawning area to which swordfish should migrate annually (thereby increasing the probability that swordfish might be caught), productivity and longevity of the trap deployment for the target species (bluefin tuna), and completeness of the data records (e. g., number of years when records were available, presence of additional data such as dates and sizes of the catch). The spawning time for swordfish in the
Mediterranean, and in particular in the Tyrrhenian Sea is June-July (Romeo et al., 2009b), and therefore overlaps with that of bluefin tuna in the same region (June-July; (Mather et al., 1995)).  The timing of tuna trap deployment therefore is not only suitable for capturing migratory bluefin tuna, but hypothetically also for swordfish.

The sites chosen for our investigation were on Sicily (Milazzo; Favigna and Formica, hereafter referred to as F+F; Capo Passero; Fig. 1) and Sardinia (Portoscuso).  The three Sicilian locations cover the northeast, southeast and western corners of
the triangular-shaped island, and trace the potential migration route of swordfish to and from the spawning area.  We selected a trap location in Sardinia (Portoscuso) because this is the only bluefin tuna trap still active in the entire

Mediterranean Sea and gives us the opportunity to study dynamics of the species in the most recent time period. The four traps therefore cover a relatively wide area approaching a major spawning area in the Mediterranean Sea (Abid and Idrissi, 2006; Romeo et al., 2009b). As a result, the traps and the spawning area are potentially visited by swordfish from a large

part of the Mediterranean Sea (i. e., immigrating from both within and outside Italian and EU waters towards this spawning area).

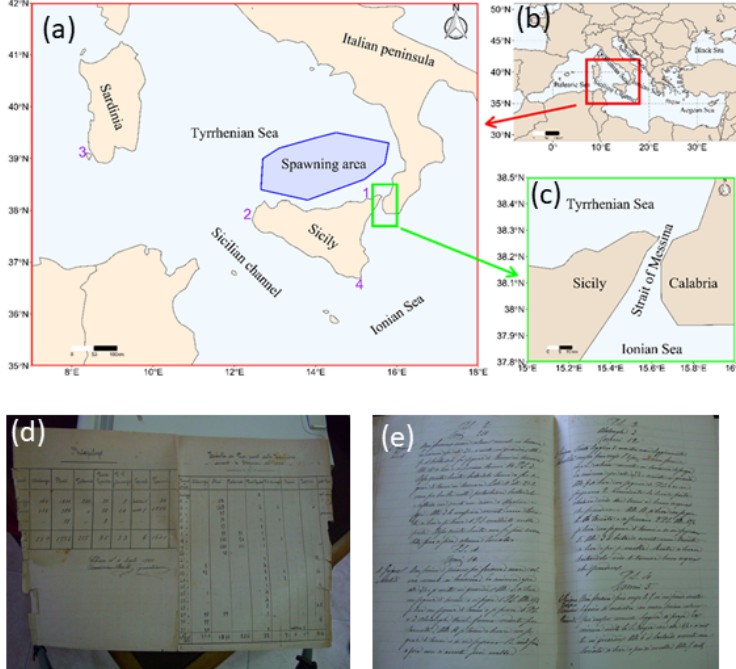

Figure 1. Panels a-c: Maps showing locations of bluefin tuna traps whose swordfish bycatch data were retrieved from historical fishing reports and accounts and geographic features mentioned in the text. Blue area on map in panel (a) indicates the

approximate location of the swordfish spawning area. Photos below (panels d and e) the maps show examples of pages from the fishing record books for the Milazzo trap.

The trap location at Favignana and Formica contains two traps located ca. 10 km from each other. There were separate catch records in numbers for each trap for many but not all years; for example, during the 1970s, the records do not specify from

which trap the catches were made but only present the sum caught (in numbers and weight) by the two traps. It is not possible to identify which trap caught the swordfish for such years. Inspection of the raw time series of catches for each trap showed that the traps were nearly always checked or emptied on separate days (i. e., there were < 10 days when swordfish were reported to have been caught on the same day at both traps; however, on those days, only trap-specific catch numbers, but not weights, were presented.) The compiled dataset includes the data separated by site where this is possible; in cases

where the site-specific data is not available, the sum is given under the site name "FavignanaFormica" in the dataset.

We extracted data from historical reports, archives and company accounts for the 4 traps (Fig. 1). The data extracted were the catches in numbers and/or total weights on each day when the trap was emptied during the fishing season. Traps were usually deployed from approximately April – late August or early September but varied depending on year and location. Once deployed, traps were usually emptied 1-2 times per week, with frequency increasing during the summer to 2-3 times

per week. The swordfish data we extracted are therefore bycaught individuals.

The traps used in Italy were similar to those deployed in other Mediterranean and east Atlantic tuna trap fisheries (Addis et al., 2008; Ravier and Fromentin, 2001). Those which are the basis for this investigation are similar in size and mesh size (Table 1); neither of these properties have changed during the period of our study (Addis et al., 2008). When swordfish were caught as bycatch, the numbers and/or weights were recorded.


2.3 Recovery and compilation of data (catches in numbers and weights; fishing effort; fishing dates):

The data were available in company records and logbooks. The data were handwritten for each catch day and we copied and entered the data into spreadsheet files for subsequent data analysis and interpretation. Examples of the handwritten documentation from two of the record books are shown in Fig. 1.

We extracted the number and weight of swordfish caught for all days available in these records. This enabled us to produce a catch time series for each year for each site. An annual time series consisted of the catches in numbers and or weight on each day during a given year's fishing season.

### 3 Results

The reports and documents contained bycatch data in numbers and/or weights for many years and usually resolved at the daily
time scale (i. e., catches made on single days throughout the fishing season). The temporal coverages (start and end years) only partly overlapped among the four trap sites (Fig. 2). Highest inter-annual overlap was with the Milazzo and F+F traps; in contrast the Portoscuso trap had little overlap with any other trap. Seasonal coverage was similar for the three traps at Milazzo, F+F and Portoscuso; the trap at Capo Passero was deployed and operated later in the summer than the three other traps (Fig. 2).

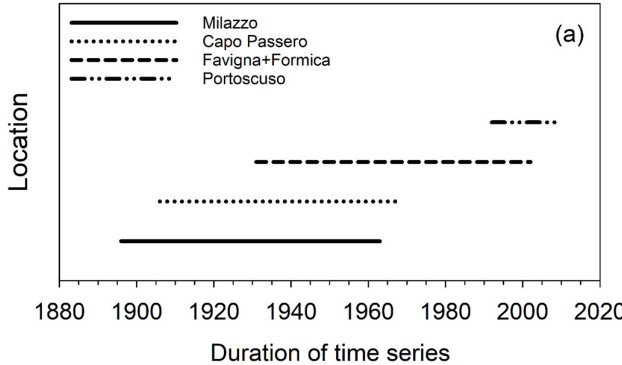

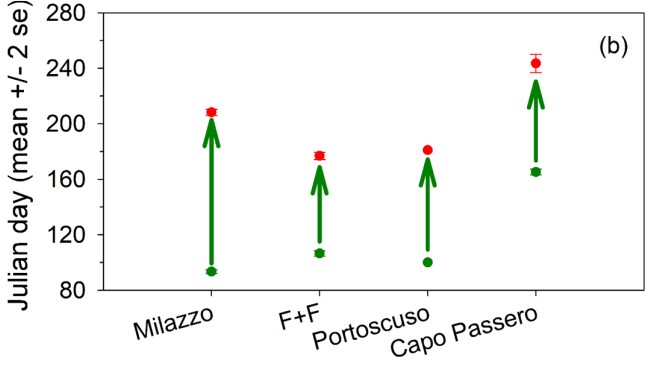


Figure 2. Summary information for temporal data availability of swordfish caught as bycatch in bluefin tuna traps in Italy. (a) years for which data are available (including gaps); (b) the mean (± 2 standard errors) start and end dates of fishing within a year at each of the four sites where traps were deployed.

Metadata for the recovered data are presented in Table 2.

Table 2. Summary of swordfish bycatch and size data recovered from 4 bluefin traps in Italy.

| Property | Milazzo | F+F | Portoscuso | Capo Passero | All sites |
|---|---|---|---|---|---|
| Time period of data available (including missing years) | 1896-1963 | 1931-2002 | 1992-2010 | 1906-1968 | 1896-2010 |
| Number of years with data available | 43 | 55 | 14 | 19 | 114 |
| Number of daily catch records (nos. and group or individual weights) | 4706 | 1016 | 743 | 140 | 6605 |
| Total number of swordfish caught | 5119 | 1670 | 744 | 652 | 8185 |
| Mean start and end date of fishing season; mean duration in days | 98, 208; 115 | 106, 177; 70 | 100, 181; 81 | 165, 244; 78 | 98, 244; 86 |
| Number of individually-weighed swordfish | 0 | 706 | 815 | 71 | 1592 |

Weight data were available for most traps but at various levels of resolution and for differing numbers of years. In one trap (Portoscuso), each individually captured swordfish was weighed. In other locations (e. g., Favignana, Formica, Capo Passero), only group weights with numbers caught for each day were available; we used these data to calculate a daily mean weight. In another site (Milazzo), group weights and group numbers were available by month, and only for a small number of years (1896-1901) (MacKenzie et al., 2021a); no weight data at any level of aggregation was available after 1901 for this site.

The data have been compiled into three datasets (see Section 5). The largest dataset contains the daily-recorded catches in numbers and/or weights for all four trap sites (MacKenzie et al., 2021c). Individual data records in this dataset consist of the trap location and relevant fishing and catch information. This includes the date and location (name and latitude-longitude coordinates) of the catch, how many swordfish were caught and in some instances how much they weighed. In some cases, the individual swordfish was weighed, but in most instances, only a group weight is available on a given fishing day. However, even at sites where only group weights were measured, there were many days when only single swordfish were caught. As a result, it was possible to compile and assemble a data set of 1592 individually-weighed swordfish during the time period

covered by these trap data (Table 2); further details of the individual weights and their variability are presented elsewhere (MacKenzie et al., 2021b).


Figure 3 shows an example time series of catches during a single year from one of the sites. These can be summed to derive annual total catches. In most locations and for most years, the date of first and last gear deployments were available and have been stored in a separate database (MacKenzie et al., 2021d); these dates can be used to evaluate whether length of

fishing season influenced the annual numbers of swordfish caught within each location and across locations throughout the time period of our study (1896-2010). The duration of fishing season and the catch data allows calculation of indices of annual catch-per-unit-effort (CPUE) in the number (or biomass) of individuals caught per day during the fishing season. The inter-annual variability in the cpue can be compared among years and sites. The data among sites can be combined into integrated indicators of abundance (MacKenzie et al., 2021b).

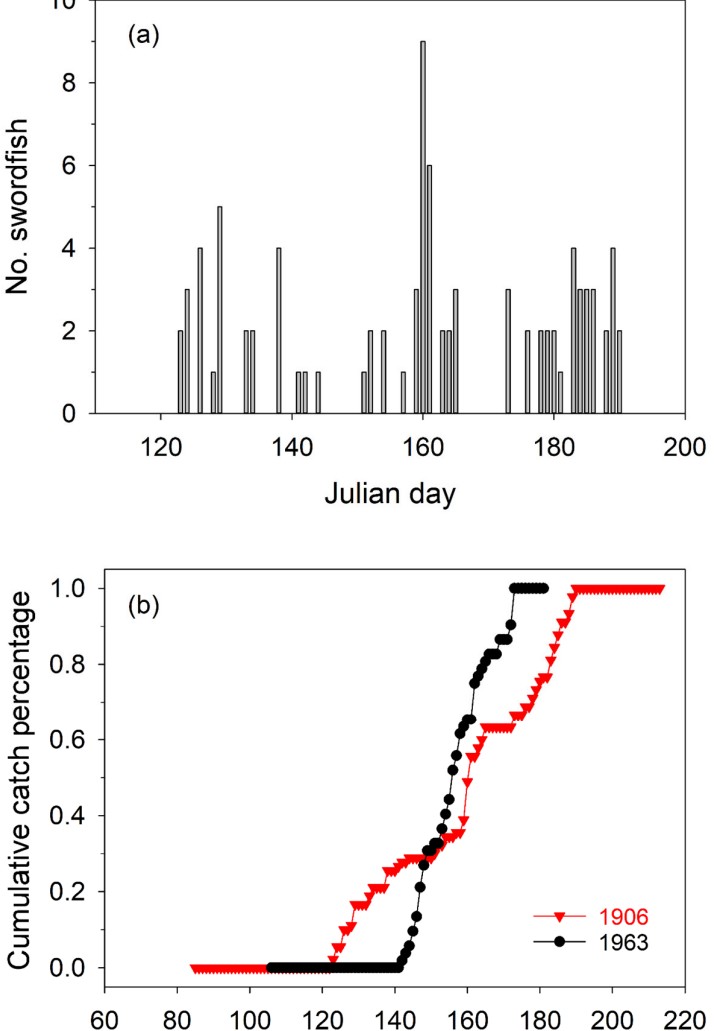


Figure 3. Examples of swordfish bycatch data extracted from bluefin tuna trap recrods. These data are from the trap located in Milazzo (see location in Figure 1). (a): bycatches on each day during the fishing season of 1906; (b): the cumulative catch percentage throughout the fishing season of 1906 and for comparison with 1963. Similar data are available in the recovered data for a total of 43 years (see also Table 2 and data files stored at Pangea.de).

Most of the swordfish caught in these traps were 20-40 kg, although several larger individuals were commonly caught (MacKenzie et al., 2021b).  Some representative sizes of individually-caught swordfish are shown in Fig. 4.

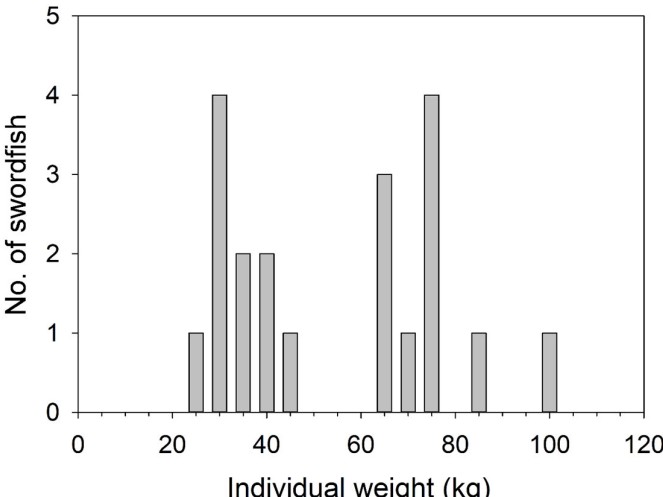

Figure 4.  Example of individual weights of swordfish caught as bycatch in bluefin tuna traps located at Favignana + Formica, Italy.  These data are for the 20 swordfish caught on 20 separate fishing days in 1946.  Similar data are available for a total of 1592 swordfish caught during 1931-2010 (see also Table 2 and online data files at Pangea.de).

**4 Discussion**

These newly-recovered historical data represent catches made by the same gear over many years and in different locations. As described above and in literature, the similarity of catch method facilitates comparisons over time and space (Addis et al., 2008; Ravier and Fromentin, 2001).

The data we have recovered can potentially be used to address at least three aspects of swordfish ecology: abundance, body size, seasonal occurrence (phenology) in local waters. Here we show some examples of the data for each of these categories and how they could potentially be used in subsequent analyses.

The raw time series of catch as displayed illustrates how many are caught on a single day, and the likelihood that individual or groups of swordfish are caught on single days.  As is evident from this example (and in the other years and locations), it is 240 clear that catches often involve single individuals, although there are days when multiple specimens are caught.  The incidence of single-capture days reflects the known solitary behaviour of swordfish as they approach spawning time and areas, as also seen from artisanal harpoon fisheries in nearby areas (i. e., Strait of Messina; (Battaglia et al., 2018; Romeo et al., 2009b).  Furthermore, it is possible that the sequence of days without catches (i. e., days between catch events) could be indicative of the spatial distribution and inter-fish distances at sea; these issues need further exploration.

The size data from the traps can be used to derive ages from growth relationships in the literature (Abid and Idrissi, 2006; Tserpes and Tsimenedes, 1995), and when compared with the known size- or age-at-maturity schedules for Mediterranean Sea swordfish (Abid and Idrissi, 2006; ICCAT, 2020), indicate the likelihood that the traps capture adults or juveniles.

The size data contained in the fishery records are among the oldest quantitative data available for Mediterranean swordfish and extend into a time period when exploitation was lower than at present. Given that mean body size in exploited fish

populations often declines when populations become overexploited (Hilborn and Walters, 2001; Jennings, 2007), new time series of mean body size that extend into low-exploitation periods can also potentially be useful indicators of changes in population status.

The high temporal resolution of the catch data allows derivation and investigation of annual catch phenologies. Although the swordfish catch data reported here are bycatches from gears deployed to catch a different species (bluefin tuna), they

may nevertheless reflect the presence of swordfish and its migration behaviour in the region and could be a basis for comparison of phenologies derived from other data sources (e. g, commercial gear such as longlines or harpoons which targets swordfish; tagging data). Examples of catch phenologies from the trap data are shown for illustrative purposes for single years for one of the sites (Fig. 3).

Numerous studies have shown how historical data such as those recovered and assembled here can provide new

understanding of the magnitudes, timing and causes of temporal variability in marine ecological properties (Engelhard et al., 2016; Lotze et al., 2014; Thurstan et al., 2015). This understanding requires acknowledgement of both the challenges and opportunities associated with such data. For example, many historical data sets have limitations which constrain the kinds of research questions which can be addressed. These limitations typically include gaps and other changes in time-space coverage due to loss of records, interruptions in the original recordkeeping practices or changes in data collection

methodologies over time (Engelhard et al., 2016; MacKenzie and Ojaveer, 2018). In addition, the data were often recorded for different purposes (e. g., economic or societal development) than those of interest to 21st century marine ecologists (Caswell et al., 2020; Lotze and McClenachan, 2013; Schwerdtner Máñez et al., 2014). As a result, some data which might appear to be obvious for an ecologist to record (e. g., full species composition and individual sizes of caught fish) may not have been prioritized by trap owners or recordkeeping authorities in the past. In addition, some details of data recording may

be lacking or the data may be aggregated at scales (e. g., only available as annual sums or group weights) that prevent investigation of some hypotheses. However, and despite limitations such as these, a growing marine historical ecology literature, some of which is cited in this report, shows that surviving historical data are often sufficient to provide new scientific knowledge at specific time-space scales or for specific ecological processes involving specific species or entire communities. As shown here, the data available in trap records can be relatively abundant, and can potentially generate new

knowledge of the ecology of particular species for time scales currently inaccessible by other data sources.

In general, the locations of the traps investigated in this study could potentially provide new perspectives to the reproductive and migration biology of the Mediterranean swordfish. Swordfish migrate to the southern Tyrrhenian Sea, and therefore swim past these trap locations, on the way to or from spawning in June-July (Perzia et al., 2016; Romeo et al., 2009b). Comparisons of the timing of annual catches across years could show associations with different factors affecting gonadal

development, including food consumption rates and temperatures. Presently, however, the role and magnitude of such effects on swordfish reproductive ecology are not documented. However, in other species (e. g., herring *Clupea harengus* Linnaeus 1758, cod *Gadus morhua* Linneaus 1758), timing of reproduction and migration to spawning areas is sensitive to the amounts and quality of prey consumed, adult condition and cumulative temperature experience when gonads are developing (Kjesbu et al., 2014; Ndjaula et al., 2010; Neuheimer et al., 2018). Recovery and analysis of historical data such

as those presented here would provide new estimates of how variable such migrations have been in the past. New dedicated

studies of gonadal development and thermal experience (e. g., via data-storage tag deployments; detailed analysis of catch data from commercial fisheries taking account of gear deployment depth, location and timing) could help fill these knowledge gaps. This knowledge could increase our understanding of the sensitivity of Mediterranean swordfish to future temperature changes caused by climate change and natural varibility.

Furthermore, historical trap data could be combined with more contemporary catch data (e. g., from longlines or harpoons (Perzia et al., 2016; Tserpes et al., 2008)) including other Mediterranean swordfish fisheries to investigate spatial aspects of swordfish reproductive and migratory behaviour. For example, swordfish also spawn in the eastern Mediterranean (Tserpes et al., 2008), and the presence of multiple spawning sites could be associated with different sub-populations within the Mediteranean Sea, especially if individuals possess migration fidelity to specific spawning sites (Romeo et al., 2009b).

However, the locations of feeding areas and overwintering sites used by potentially distinct spawning groups, and the possibility that individuals from multiple spawning sites use common feeding areas (as is the case in other pelagic species such as bluefin tuna and Atlantic herring (Dickey-Collas et al., 2010; Jansen et al., 2021; Rodríguez-Ezpeleta et al., 2019; Rooker et al., 2019) are unknown and remain to be described for Mediterranean swordfish. New analyses of multiple sources of catch data having broader spatial-temporal coverage, supplemented with new tagging studies, could be used to

investigate reproductive and migratory processes such as these.The taxonomic identify of species named in historical documents can often be difficult to assess due to multiple names for the same species used in the same and different locations or time periods; furthermore many species can have similar external appearances potentially increasing difficulties for correct species identification. In this study, there is little or no possibility for taxonomic confusion of this species with other species that could have been caught. This species is one of the world's most recognizable and famous fish species

(Stergiou, 2017) and because of its unusual appearance, is unlikely to have been mistaken by professional fishermen for other species and vice versa. Another billfish, Mediterranean spearfish, *Tetrapturus belone* Rafinesque 1810, is also present in the area (Romeo et al., 2009a, 2015) but its appearance is sufficiently different from swordfish that the two species are easily distinguishable. Furthermore, there have been no quota or minimum size restrictions for swordfish in the Mediterranean Sea until 2017 and 2011 when regulations on TAC and minimum size respectively were implemented

(ICCAT, 2011, 2017). Consequently, there has been little incentive for fishermen to misreport their catches, especially before these years. As a result, the written records are highly likely to be true representations of the amounts of swordfish caught by the traps.

Our finding that the bluefin tuna trap records regularly contain swordfish bycatch in amounts that could increase ecological

understanding are consistent with some other data sources. Analysis of the species composition of a smaller trap located on the northern Tyrrhenian Sea coast of Italy indicated that swordfish were sometimes caught as bycatch at this location (Britten et al., 2014). In addition, swordfish have been caught as bycatch in a Spanish bluefin tuna trap at Barbate (Gulf of Cadiz; (Rodríguez-Roda, 1964). The data recovered in our study suggest that there may be potentially many more data in other trap logbooks and accounts for other locations and time periods than we have investigated here. These deserve investigation in

future.

**5 Data availability**

The data are deposited in three datasets in the Pangaea open-access online repository at the following temporary links (permanent links and DOIs from Pangaea to follow):

https://www.pangaea.de/tok/8d1a24cfe1e5b9f79f61f40eaf8f804e469873f0 (MacKenzie et al., 2021c)

https://www.pangaea.de/tok/b7e606c35804776223b2f6f372061ac212458a0d (MacKenzie et al., 2021a)

The first dataset contains the bycatch data in numbers and/or weight on given days during 1896-2010 for the four traps. The second dataset contains monthly-resolved bycatch data in weights for the Milazzo trap for the years 1896-1901. The third dataset contains the estimated start and end dates and durations for the annual fishing seasons in each of the four trap sites.

## 6 Conclusions

The historical reports and archives we have investigated contain new quantitative information that can be used to make new analyses and discoveries of swordfish ecology in the Mediterranean Sea. The recovered data document swordfish caught as bycatch using similar methods over time and among sites, thereby allowing a high degree of comparability across time and space. The datasets allow investigations of multi-decadal variability in swordfish ecology in the Mediterranean Sea during most of the 20th century, including time periods for which quantitative data are rare or non-existent, and can potentially establish new baseline knowledge for this population.


## 7 Code Availability

No code was used or developed in this study.

## 8 Author contributions

BRM and TR designed research. BRM analysed data, prepared figures, wrote first draft of the manuscript and edited later versions. TR, PA, PB, PC, FA and GS assembled data, assisted with interpretations and edited and reviewed the manuscript.

## 9 Competing interests

The authors declare that they have no conflict of interest.


## 10 Acknowledgements

We thank Prince Belmonte for allowing access to and recovery of original catch data from records of the trap located at Capo Passero, Sicily. This work was supported by the participating institutes and is a contribution to the ICES Working Group on History of Fish and Fisheries and the Oceans Past Platform; we thank participants in this group and network for 355 discussions and feedback; BRM especially thanks Saša Raicevich and Henn Ojaveer. An initial travel grant (Short Term Scientific Mission) was provided to BRM from the EU COST Action IS1403 (Oceans Past Platform: OPP) to visit colleagues (TR, PB, PC, FA) at Milazzo, Italy to discuss and plan project objectives and data recovery. We thank the reviewers and editors for their input and suggestions to the manuscript.

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
