# Peer review of "New historical data for long-term swordfish ecological studies in the Mediterranean Sea"

_Earth System Science Data, 2021_

## Author Response (AR1)

Reply of authors:

**Comment from Reviewer 1:**

It is highly interesting and timely paper tthat contributes at the same time to the knowledge about the populations of a marine species target of long term exploitation and which status is not fully addressed, while showing the value of alternatives sources of information besides current quantitive assessments. Historical marine ecology and the resourcing to long term data series brings new and relevant information and can directly contribute to the conseravtion and/or management of marine fish populations. Moreover, it opens new avenues of research that complement current day research in fisheries' management with historic previously neglected data while addressing long term interactions and impacts of societies in marine ecosystems. I strongly recommend the publication.

**Response of authors to Reviewer 1:**

We thank the reviewer for her/his comments and appreciation of the merits of the manuscript. As the reviewer had no requests or suggestions for changes, we have not edited the manuscript to address her/his comments.

**Comment from Reviewer 2.**

In my opinion, the extracted numbers and weights of swordfish from the process of data recovery are very poor. Despite the authors declared an assembly of 1592 individually weighted swordfish, I do not see any length or weight frequency distributions neither by time nor by site. It should be interesting to see how the structure in size of the swordfish catches were distributed over time and space, especially considering potential swordfish migratory pattern and the exceptional case of the Capo Passero tuna trap, which was a return-trap.

Regarding the period considered, I think that it is not valuable since:

a) time series from Milazzo is unreliable because from 1901 "no weight data at any level of aggregation was available" (line 192).

b) data from F+F are only grouped data of weights and numbers. Despite the authors declare that a daily mean weight is available for those locations for 14 years (1931-2002), for 1 fish a day and for 743 days of captures, I see a frequency distribution of only of 20 specimens in Figure 4 and they were caught only in 1946 (line 223).

c) regarding Milazzo, the authors reported only data from fishing season of 1906 (line 216) and the year cumulative catch percentage compared to that of 1963 (line 217). I do not see how these data can be used to help the improvement of the actual knowledge on the swordfish population in the Mediterranean Sea.

Therefore, I do not understand how the poor data presented can be a sufficient basis to improve the swordfish management decisions (line 33) regarding the recovery of the species. Time series used by the authors continue to be insufficient to allow any diagnosis and attribution of changes in swordfish population dynamics.

It should be very interesting to try to focalise the observation on the period that provide the best swordfish data set, in terms of amount of individual weights by trap and time to improve the knowledge on the structure of the historical swordfish catches.

Finally, there is an important lack of papers regarding the reproduction biology and population dynamics, migratory pattern and structure in the bibliography section.

Swordfish should be cited as Latin name Xiphias gladius (L. 1758), at least when it is cited for the first time (line 48).

For all the above-mentioned reasons, I feel that the manuscript is not suitable for publication, unless it is deeply improved in order to provide any improvement in the knowledge of this species and, overall, any new ecological knowledge regarding the swordfish population dynamics in the Mediterranean Sea.

**Response of authors to Reviewer 2**:

We thank the reviewer for her/his comments on our manuscript. We believe that the dialogue with referees is always a source of reflection and it is a key step of the scientific process. However we are a bit worried - in the present case - that Reviewer 2 was slightly unfamiliar with the scope and format of manuscripts published by this journal.

This Journal publishes manuscripts whose approach, philosophy and style is very different from those of common standard research articles. We appreciated many of the suggestions by Reviewer 2 although some scope-related comments are not fully applicable to the specific case of our manuscript. When preparing this data manuscript, we followed journal guidelines and scope - the description of why and how data were collected, where, when, what methodologies were used, and some perspectives on how they could be used in future. In addition, to corroborate these aspects, the raw data themselves were submitted.

Specifically, our manuscript followed these criteria and we provided both the full datasets (as is mandatory for submission of a manuscript to ESSD), and some illustrational examples of the data which show what they look like and how they can be used. We have therefore focussed our manuscript and its disscussion on the data themselves, rather than on ecological interpretation of trends or variations in these data sets as expected and requested by the reviewer. Such topics instead fit the scope of other interpretation-focussed journals (e. g., Marine Ecology Progress Series, Global Change Biology, ICES Journal of Marine Science).

Another criticism raised by this reviewer was the temporal extension of the datasets, probably based on a misunderstanding of what was shown in some of the figures. This comment is based on an incorrect understanding of our dataset, i. e., the reviewer has concluded that our manuscript presents only a very short and small amount of data, apparently overlooking the meta-data summarized in our Table 2 and not realizing that our figures only showed *examples* of some of the data recovered. We now realize that we

were not clear enough in our text or figure captions. Consequently, in the new version, we tried to be more precise in saying that our time series covered many decades (please look at Table 2) and as also shown clearly in the raw datafiles submitted with the manuscript. We now state in the captions to Figures 3 and 4 that the entire datasets are summarized in Table 2 and available online; this text is now in addition to the original wording that stated that these figures show examples of some of the data).

Further, the criticism by Reviewer 2 of the length and quantity of our datasets was, among other things, in conflict with what the two other reviewers noticed about the temporal extension of dataset and the overall quality of the manuscript. The other reviewers were aware that the coverage of our datasets was several decades long and not only a few years as commented by Reviewer 2.

We believe therefore that an unfortunate misunderstanding of the specific scope and content of a data manuscript led to false expectations by the Reviewer 2. The approach adopted in our manuscript is consistent with the scope of manuscripts submitted and published by this journal, and we believe that the way we presented and described the data are consistent with its standards.

However, we appreciated the reviewer's interest in how the data could be used to increase understanding of the biology of the stock, how the data could be affected by reproductive biology, reasons for variations etc. The increase in ecological knowledge of the species in the region will come later in other manuscripts by ourselves and hopefully by others in the community. Nevertheless, if we include such analyses and interpretations in this manuscript we would have been outside the journal scope.

Detailed responses (our comments are preceded with two asterisks ** and appear in *red italics*) are provided below:

Reviewer 2:

In my opinion, the extracted numbers and weights of swordfish from the process of data recovery are very poor. Despite the authors declared an assembly of 1592 individually weighted swordfish, I do not see any length or weight frequency distributions neither by time nor by site. It should be interesting to see how the structure in size of the swordfish catches were distributed over time and space, especially considering potential swordfish migratory pattern and the exceptional case of the Capo Passero tuna trap, which was a return-trap.

*\*\*We provide new time series covering many decades. The meta-data are presented in Table 2, including samples sizes, and the coverage of years. This clearly shows that there are many more data than what the reviewer has stated in his/her comments. Furthermore, the complete datasets are presented as assets to this manuscript and the duration and coverage is evident from those files.*

Regarding the period considered, I think that it is not valuable since:

1. a) time series from Milazzo is unreliable because from 1901 "no weight data at any level of aggregation was available" (line 192).

   *\*\*The time series (of group weights) is very short, only 1896-1901, and this was stated in the manuscript. The data for those years were available in the historical reports, and are reliable, so we recovered them for the community. We disagree with the reviewer's comment.*

2. b) data from F+F are only grouped data of weights and numbers. Despite the authors declare that a daily mean weight is available for those locations for 14 years (1931-2002), for 1 fish a day and for 743 days of captures, I see a frequency distribution of only of 20 specimens in Figure 4 and they were caught only in 1946 (line 223).

*In this data manuscript, and in this figure in particular, we show only an example of the type of data available. The full dataset contains 100s of individually-weighed swordfish as stated in the meta-data Table 2, and as can be seen in the accompanying data set that was submitted with the text manuscript. A full analysis of the time-space variability of these weights is not suitable for this manuscript and more appropriate for a separate manuscript where the data are more fully interpreted ecologically.*

3. c) regarding Milazzo, the authors reported only data from fishing season of 1906 (line 216) and the year cumulative catch percentage compared to that of 1963 (line 217). I do not see how these data can be used to help the improvement of the actual knowledge on the swordfish population in the Mediterranean Sea.

*This comment is similar as previous ones which have criticized the data we have recovered and presented. We have these kinds of data for 43 years between 1896 and 1963 (stated in Table 2, and therefore not only the two years shown in the figure and referred to by the reviewer. We only show examples from two years to display the type of seasonal variations evident in different years and the potential for new analyses based on these kinds of historical data. Again, the full catch data by date dataset was available to the reviewer and was part of the submission materials for this journal. We believe there has been an oversight and misunderstanding by the reviewer which has led to these comments.*

Therefore, I do not understand how the poor data presented can be a sufficient basis to improve the swordfish management decisions (line 33) regarding the recovery of the species. Time series used by the authors continue to be insufficient to allow any diagnosis and attribution of changes in swordfish population dynamics.

*Actually we provided many more data than we show in the figures and we are aware that this comment provided by Reviewer 2 was based on our lack of clarity. Nevertheless, in Table 2 we reported meta-data and we submitted data files. We emphasize that we have only presented examples of some of the kinds of data and analyses which are possible with these new datasets. We have added some text lines about the general challenges of recovering, analysing and interpreting historical data to the Discussion at lines 259-275.*

It should be very interesting to try to focalise the observation on the period that provide the best swordfish data set, in terms of amount of individual weights by trap and time to improve the knowledge on the structure of the historical swordfish catches.

*Agree. Indeed the entire time period covered by our study will be able to improve knowledge of historical swordfish catches.*

Finally, there is an important lack of papers regarding the reproduction biology and population dynamics, migratory pattern and structure in the bibliography section.

*We have added several lines of text and references to the Discussion about these biological aspects. Please see lines 276-300 in the track-changes version of the manuscript.*

Swordfish should be cited as Latin name Xiphias gladius (L. 1758), at least when it is cited for the first time (line 48).

*\*\*Done, and also done for names of other species mentioned in the manuscript.  Please see lines 19, 20 and 282 in the track-changes version of the manuscript.*

For all the above-mentioned reasons, I feel that the manuscript is not suitable for publication, unless it is deeply improved in order to provide any improvement in the knowledge of this species and, overall, any new ecological knowledge regarding the swordfish population dynamics in the Mediterranean Sea.

*\*\*As stated above, expectations by the reviewer - even though legitimate - fell outside the Journal scope. With no aim to be repetitive, this manuscript is a data paper and focused on the description of new data sets – how the data were obtained, recovered, and now made available to the scientific community. We have provided some  biological background to support the justification for their recovery.  However, any further analysis and interpretation to  expand - in this specific data-paper - the theoretical ecological context is outside  Journal scope.*

**Authors' changes in manuscript to Reviewer 2:**

We now state in the captions to Figures 3 and 4 that the entire datasets are summarized in Table 2 and available online; this text is now in addition to the original wording that stated that these figures show examples of some of the data.  Please see lines 219-220 and 226-227 of the track-changes version of the manuscript.

We have added some text lines about the general challenges of recovering, analysing and interpreting historical data to the Discussion at lines 259-275.

We have added several lines of text and references to the Discussion about the biological aspects mentioned by the reviewer. Please see lines 276-300 in the track-changes version of the manuscript; new cited references are included in the bibliography.

Regarding the inclusion of the taxonomist's name for the identified species: we have added Linnaeus to the species names at lines 19, 20 and 282 in the track-changes version of the manuscript.

**Comment from Reviewer 3:**

The paper provides some historical data series of swordfish catches in the central Mediterranean that would help to improve our knowledge regarding changes on the size structure and abundance of the stock over time. Given that the data refer to a low exploitation period, certain data metrics (e.g. mean weight of individual fish) could be potentially useful indicators of population changes induced by intense fishing activities in the most recent years. In this concept, data provide complementary information that can be utilized for conservation purposes and support the efforts for rebuilding the over-exploited Mediterranean swordfish stock, through appropriate management measures. Based on the above, I recommend publication of the article.

**Response of authors to Reviewer 3:**

We thank the reviewer for her/his comments and appreciation of the merits of the manuscript. As the reviewer had no requests or suggestions for changes, we have not edited the manuscript to address her/his comments.

[revised manuscript text omitted]